# Characterization of Microstructures and Fatigue Properties for Dual-Phase Pipeline Steels by Gleeble Simulation of Heat-Affected Zone

**DOI:** 10.3390/ma12121989

**Published:** 2019-06-20

**Authors:** Zuopeng Zhao, Pengfei Xu, Hongxia Cheng, Jili Miao, Furen Xiao

**Affiliations:** 1Jiangsu Key Laboratory of Coast Ocean Resources Development and Environment Security, Hohai University, Nanjing 210098, China; zpzhao@hhu.edu.cn (Z.Z.); chenghongxia@hhu.edu.cn (H.C.); 2Jiangnan Shipyard (Group) Co. LTD., Shanghai 201913, China; cruisemiao@hotmail.com; 3Key Lab of Metastable Materials Science & Technology and College of Materials Science & Engineering, Yanshan University, Qinhuangdao 066004, China

**Keywords:** thermal cycle simulation, dual-phase steel, heat-affected zone, fatigue property

## Abstract

To increase transmission efficiency and reduce operation cost, dual-phase (DP) steels have been considered for pipeline applications. Welding has to be involved in such applications, which would cause a localized alteration of materials and cause many potential fatigue issues to arise under cyclic loading. In this work, the fatigue crack propagation and fatigue life of simulated heat-affected zone (HAZ) were examined. Results indicate that when the maximum stress is at the same magnitude, the fatigue life at a peak temperature of 1050 °C is very close to that of a peak temperature of 850 °C, and both of them are higher than that of a peak temperature of 1350 °C. The changes in *da/dN* with Δ*K* for HAZ subregions are attributed to the variation of crack path and fracture mode during the crack propagation. The fatigue cracks may propagate along the bainite lath preferentially in coarse-grained HAZ (CGHAZ), and the prior austenite grain boundaries can change the crack growth direction. A considerable amount of highly misoriented grain boundaries in fine-grained HAZ (FGHAZ) and intercritical-grained HAZ (ICHAZ) increase the crack growth resistance. The difference of fatigue crack propagation behavior in HAZ subregions between actual and simulated welded joints was also discussed.

## 1. Introduction

Dual-phase (DP) steel, a type of advanced high-strength steel (AHSS), is being increasingly used in the transportation of oil and natural gas due to its good processability and balanced combination of strength and ductility [1,2,3]. This excellent performance is attributed to dual-phase microstructural characteristics. For the ferrite/bainite DP steel, the ferrite as a soft phase have higher ductility, and the bainite as a hard phase is strong [3,4]. A high-strength pipeline steel with excellent deformability is obtained. However, a microstructure evolution will occur in and around the welded area during the welding process, which leads to the deterioration of performance in welded joints [5,6].

From a metallurgical point of view, welding can be identified as a thermal cycle that consists of continuous heating to a certain peak temperature and cooling to room temperature [7]. The weld thermal cycle changes the base metal and results in an inhomogeneous microstructure in heat-affected zone (HAZ) [8]. The previous research reported that HAZ is often the most critical region with regard to potential fatigue failures, and this phenomenon is closely related to their microstructures [6,9,10]. The evaluation and prediction of fatigue behavior are very important to avoid catastrophic failure particularly in welded DP steel structures [11,12]. Therefore, the fatigue properties of HAZ for DP steel should be given more attention.

In most previous studies, the welded joint specimens were usually used to study the effect of microstructures of HAZ on fatigue behavior [9,10,12]. However, research of the HAZ in actual welded joints is not easy due to the narrowness of the HAZ [13]. As a suitable method to prepare specimens, welding thermal simulation was used to investigate and identify the various HAZ subregions [14,15]. Hutchinson et al. used simulation to show the relationships between HAZ microstructures and toughness [16]. Boumerzoug et al. investigated the effect of heat treatment obtained by thermal cycles simulation on low carbon steel microstructure and indicated that microstructures correspond exactly to those observed in actual welded joint [13].

In this paper, the fatigue life and crack propagation behaviors of simulated HAZ for DP pipeline steel were investigated. The difference of fatigue crack propagation behavior in HAZ subregions between actual and simulated welded joints was also discussed. This study provided beneficial experimental data for safe designs of gas transmission pipelines.

## 2. Materials and Methods 

In this work, a commercial ferrite/bainite DP X80 pipeline was selected. The chemical composition and microstructure characterizations of DP steel are shown in Table 1 and Figure 1a. DP steel consists of as-rolled mixed microstructure of polygonal ferrite (PF) and bainite (BF).

Thermal cycling simulations typical of the thermal cycles of the HAZ were performed using samples of 10×10×65 mm dimensions in a thermomechanical Gleeble-3800 system. The heat input of thermal cycles was 45 kJ/cm, which corresponds to the practical production. Figure 1b shows the thermal cycle procedure based on ANSYS analysis. For example, the specimen of coarse-grained heat affected zone (CGHAZ) was heated up at a constant rate of 130 °C/s to a peak temperature of 1350 °C and held for 1 s, then cooled down to 800 °C in 24 s and to 500 °C in 54 s, respectively. The other thermal cycles were performed at different peak temperatures, and the cooling speeds from 800 °C to 200 °C were the same as that in the above thermal cycle.

After the HAZ thermo-mechanical simulations, fatigue tests were performed on two types of fatigue samples that were machined from HAZ samples, as shown in Figure 2. Stripe and single-edge notched bend (SENB) specimens were used in the fatigue life test and fatigue crack propagation test, respectively.

The fatigue life tests were executed through an MTS servo-hydraulic universal testing machine (MTS 858, MTS, U.S.). The sinusoidal waveform and a stress ratio (*R* = *σ*_min_/*σ*_max_) of 0.1 were applied to specimens at a frequency of 20 Hz. The total fatigue cycles of each fatigue specimen were counted and considered as the fatigue life. The fatigue crack propagation tests were performed using an MTS 858 Mini Bionix testing machine (MTS, U.S.). The *R*-ratio and frequency were set at 0.1 and 20 Hz, respectively. The fatigue pre-cracking of each SENB specimen located in simulated zone and was generated by cyclic loading until a crack with the desired length was formed. Tensile tests were performed using an MTS 858 Mini Bionix testing machine at a strain rate of 1 × 10^−3^ s^−1^. An extensometer with a gauge length of 10 mm and a strain limit of 20% was used to measure strain during the tensile test. This procedure was conducted following the ASTM standard [17,18]. All of the above test and parameters are shown in Table 2.

The analysis of microstructures was performed by optical microscopy (OM, Axiover 200MAT, ZEISS, Germany), scanning electron microscopy (SEM, S-4800, Hitachi, Japan), and electron back-scattered diffraction (EBSD, EDAX, U.S.). The SEM and EBSD were used to observe the fracture surface and the growth characteristic, respectively.

## 3. Results

### 3.1. Microstructure and Tensile Properties

Figure 3 shows OM images of the simulated welded microstructures at different peak temperatures. In CGHAZ (1350~1150 °C), granular bainite (GB) and the prior austenite grain boundary network exist in the microstructure (Figure 3a). GB is considered that some island constituents are distributed in matrix. Due to staying at a high temperature too long, the prior austenite grain size grows heavily. With the decrease of peak temperature, the prior austenite grain size decreases (Figure 3c). As the peak temperature decreases to 1150 °C, the microstructure mainly consists of quasi-polygonal ferrite (QF) and BF. The prior austenite grain boundaries disappear at the same time, which is regarded as fine-grained heat affected zone (FGHAZ) (Figure 3d). In intercritical-grained heat affected zone (ICHAZ), the as-rolled mixed microstructures of PF and BF appear in the microstructure (Figure 3f).

Figure 4 shows yield strength, tensile strength, and uniform elongation of samples after thermal cycling simulation. With the increase of peak temperature, the yield strength increases. The same variational trend is also found in the tensile strength. When the peak temperature is 950 °C, the yield strength and the tensile strength reach a minimum value. On the contrary, the uniform elongation increases with the increase of peak temperature first, then reduces and reaches the maximum at a peak temperature of 1050 °C.

### 3.2. Fatigue Properties

The *S*−*N* curves obtained by the load control fatigue test are plotted in Figure 5. When the maximum stress is at the same magnitude, the fatigue life at a peak temperature of 1050 °C is very close to that of a peak temperature of 850 °C, and both of them are higher than that of a peak temperature of 1350 °C. The peak temperature of 1350 °C, 1050 °C, and 850 °C correspond to CGHAZ, FGHAZ, and ICHAZ, respectively. Thus, compared with other subregions, the fatigue life declines severely in CGHAZ. As described above, each HAZ subregion demonstrates many different microstructure characteristics because of different peak temperatures, especially CGHAZ (Figure 3). Therefore, the results of fatigue life test may be related to the effect of microstructure on the crack propagation behavior.

Since a fatigue pre-crack is not formed, the full fatigue life is composed of a cycle of fatigue crack initiation, propagation, and fracture in the fatigue life test. The fatigue crack growth rate (*da/dN*) that is used as a basis to predict the fatigue life was also measured in the fatigue crack growth test. Figure 6 shows the relationship of *da/dN* with a stress intensity factor of crack tip (Δ*K*) for subareas of HAZ. When Δ*K* exceeds 27 MPa∙m^1/2^, *da/dN* in CGHAZ, ICHAZ, and FGHAZ have the same tendency of changes with Δ*K* before ultimate fracture region. As Δ*K* is less than 27 MPa∙m^1/2^, *da/dN* is the maximum in CGHAZ, is higher in FGHAZ, and is the minimum in ICHAZ. At the early crack propagation stage, fatigue cracks in CGHAZ grow fast and are easy to form large-size cracks in a short time. The subsequent accelerates cracks entering into ultimate fracture stage, which may explain why the fatigue life declines severely in CGHAZ.

### 3.3. Fracture Analysis

Figure 7 shows the SEM fracture surface of the specimen at a peak temperature of 1350 °C in fatigue crack growth test. The river line patterns fanning out and flowing along the crack propagation direction are visible in early propagation region (Figure 7a). Whereas, some local river lines have different characters in direction, which may be related to orientation distributions of the bainite lath. In addition, the fracture surface features appear as the fatigue striation and fine second crack (Figure 7b). With the growth of fatigue crack, the characteristics of fracture surface differ. The spacing of fatigue striation increases, and some large second cracks can be observed (Figure 7c,d).

Figure 8 illustrates the characteristics of the fracture surface of the specimen at a peak temperature of 1050 °C. In the early propagation region, the fracture surface shows the ductile feature of flat facets and a small amount of second cracks (Figure 8a,b). As the crack continues to propagate inward, the size of flat facets and second cracks increase, and the fracture surface turns from initially rough into smooth (Figure 8c). With the increase of crack growth rate, the number of second cracks increases significantly and flake-like structures appear in the fracture surface (Figure 8d).

Figure 9 shows the SEM fracture surface of the specimen at a peak temperature of 850 °C. The fracture surface feature is similar to that of the specimen at a peak temperature of 1050 °C. The fatigue striations and second cracks are visible, and some flat facets exist near the second cracks (Figure 9a,b). With the growth of fatigue crack, the plastic deformation of flat facets is accelerated and its size increases (Figure 9c). The large second cracks perpendicular to the propagation direction exist along the boundaries of the fracture facet (Figure 9d).

## 4. Discussion

### 4.1. Effect of Microstructure on Fatigue Crack Growth Behavior

The fatigue properties and the fracture surface features show that obvious differences exist in fatigue crack growth behaviors for different HAZ subregions. This result is related to the changes in crack path and fracture mode because of the possible microstructural sensitivity of fatigue crack propagation behavior.

The fatigue life test shows that the fatigue life declines severely in CGHAZ (Figure 5). As mentioned before, this result is caused by the higher *da/dN* in CGHAZ at the early crack propagation stage. Figure 10 shows the details of crack propagation path in HAZ subregions. When the cracks across the bainite colonies, the extended direction parallels to the lath BF boundaries (in Figure 10a designated A), and a few deflections occur in the prior austenite grain boundaries (in Figure 10a designated B). This phenomenon is also in accordance with the observation that the fracture surface appears local river lines with different characters in direction (Figure 7a). Previous studies have indicated that highly misoriented grain boundaries can efficiently arrest the propagation of cleavage micro-cracks [19,20]. For CGHAZ, bainite lath boundaries are mostly low-misorientation-angle boundaries (Figure 11). Thus, the fatigue crack may propagate along the bainite lath preferentially because of a weaker crack propagation resistance. Whereas, a considerable amount of highly misoriented grain boundaries exist in ferrite-ferrite and ferrite-BF boundaries because of a mixture of fine BF and ferrite microstructures (Figure 12). As shown in Figure 13, the number fraction of highly misoriented angle in FGHAZ is far more than that in CGHAZ, which implies more crack propagation resistance in FGHAZ. That is why the crack path in FCHAZ is more tortuous and appears some branches (Figure 10b). Therefore, *da/dN* in FGHAZ reduces compared with that in CGHAZ on account of the crack path deflection that decreases the local driving force for crack propagation. 

Moreover, some micro-cracks and micro-voids appear near the main crack and most of them exist in the boundaries (Figure 12). Owning to the difference between BF and ferrite in strength, the incoordinate deformation may occur under cycle loading [3]. Slip bands are generated in the crack tip plasticity zone and accumulate against ferrite-BF boundaries [3,21]. Along with accumulative plastic strain intensifying, the micro-cracks form near the main crack, which is in accordance with the observation that numerous plastic deformations and clear second cracks exist around the flat facets (Figure 8a,b). This phenomenon also appears in ICHAZ (Figure 9 and Figure 14). The formation of these second cracks consumes a substantial energy of the main crack propagation to some extent. This is one of the reasons *da/dN* in FGHAZ and ICHAZ reduces compared with that in FGHAZ. In the case of ICHAZ, it is noteworthy that the Δ*K* and da/dN plot is more curve and seems to be a transition point. The results are similar to that described in previous studies [3,4]. 

### 4.2. Difference between Simulated and Actual Welded Joints in Fatigue Crack Growth Behavior

To compare and analyze the difference between simulated and actual welded joints in fatigue crack growth behavior, a *da/dN*-Δ*K* curve after mosaic is obtained by thermal simulation samples, as shown in Figure 15. The simulated curve consists of different stages of each *da/dN*-Δ*K* curves in HAZ subregions. The variation trends between simulated and actual crack growth curve are basically consistent with an increase of Δ*K*. The maximum growth rate of *da/dN* is in CGHAZ, diminishes up to FGHAZ, then increases to ICHAZ. The crack growth curve can be described by Paris model base on the corresponding subregions [22], given by Equation (1)
*da/dN* = *C*(Δ*K*)*^m^*,(1)
where *C* and *m* are constants with materials. The calculated *C* and *m* of different subregions in simulated and actual crack growth curve are listed in Table 3. The difference of *da/dN* and growth rate exist in different subregions. In CGHAZ, the crack growth curve of simulated samples is located below that of actual welded joint, i.e., *da/dN* of simulated samples is lower when Δ*K* at the same magnitude. This result is related to an increase of crack propagation resistance because of the changes in the size of prior austenite grain. As stated above, the prior austenite grain boundaries play an important role to arrest fatigue crack growth (Figure 10a) [19,20]. For simulated samples, the size of prior austenite grain remains substantially constant. On the contrary, the grain sizes gradually decrease with distance from the fusion line for actual welded joint [12]. As a result, the number of prior austenite grain boundary per unit area increases during the crack growth process and the resistance of fatigue crack propagation also increases, which is consistent with the variation that *da/dN* growth rate decreases with a increase of Δ*K*. In fact, the size of prior austenite grains close to the fusion line is larger than that of simulated specimen at a peak temperature of 1350 °C [23]. Therefore, *da/dN* of actual welded joint reaches a higher level at the early crack propagation stage.

In FGHAZ, the crack growth curve of actual welded joint agrees with that of simulated samples. The reason is that the microstructure has less change with the increase of peak temperature in FGHAZ because of the smaller grain (Figure 3d,e). When the fatigue crack propagates toward ICHAZ, the *m* value of simulated curve is similar with the actual crack growth curve, but *da/dN* is different. In general, although the variation trends between simulated and actual crack growth curve are basically consistent with the increase of Δ*K*, some differences of *da/dN* and its growth rate exist in different subregions. In other words, the samples obtained by thermal cycling simulation can be used to research the microstructure and the fatigue crack growth behavior of HAZ. For the propagation life prediction of actual welded joint, however, the simulated crack growth curve exists some deviations.

## 5. Conclusions

(1) When the maximum stress is at the same magnitude, the fatigue life for X80 steel at a peak temperature of 1050 °C is very close to that of a peak temperature of 850 °C, and both of them are higher than that of a peak temperature of 1350 °C.

(2) When Δ*K* exceeds 27 MPa∙m^1/2^, *da/dN* in CGHAZ, ICHAZ, and FGHAZ have the same tendency of changes with Δ*K* before ultimate fracture region. As Δ*K* is less than 27 MPa∙m^1/2^, *da/dN* is the maximum in CGHAZ, is higher in FGHAZ, and is the minimum in ICHAZ.

(3) The changes in *da/dN* with Δ*K* for subregions are attributed to the variation of the crack path and the fracture mode during fatigue crack propagation. The fatigue crack may propagate along the bainite lath preferentially in CGHAZ, and the prior austenite grain boundaries can change the crack growth direction. A considerable amount of highly misoriented grain boundaries in FGHAZ and ICHAZ increase the crack growth resistance, thus the fatigue life is developed.

(4) The samples obtained by thermal cycling simulation can be used to research the microstructure and fatigue crack growth behavior in HAZ. For the propagation life prediction of actual welded joint, however, the simulated crack growth curve exists some deviations.

## Figures and Tables

**Figure 1 materials-12-01989-f001:**
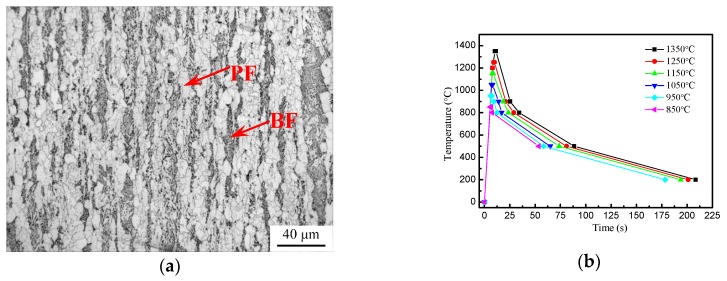
(**a**) Optical micrographs of base metal; (**b**) Thermal cycle procedure for the experiments.

**Figure 2 materials-12-01989-f002:**
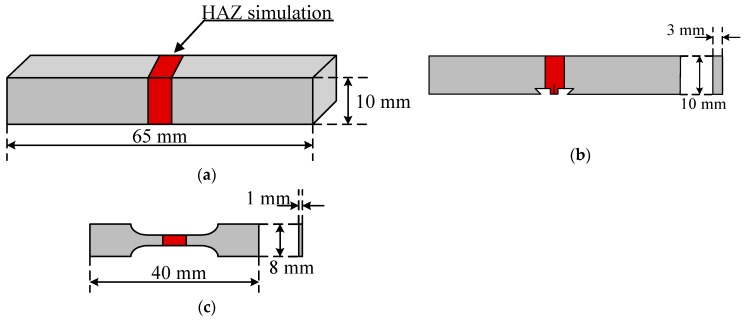
(**a**) Geometry of the thermal cycling simulations test; (**b**) Geometry of fatigue crack propagation test; (**c**) Geometry of fatigue life and tensile test specimens.

**Figure 3 materials-12-01989-f003:**
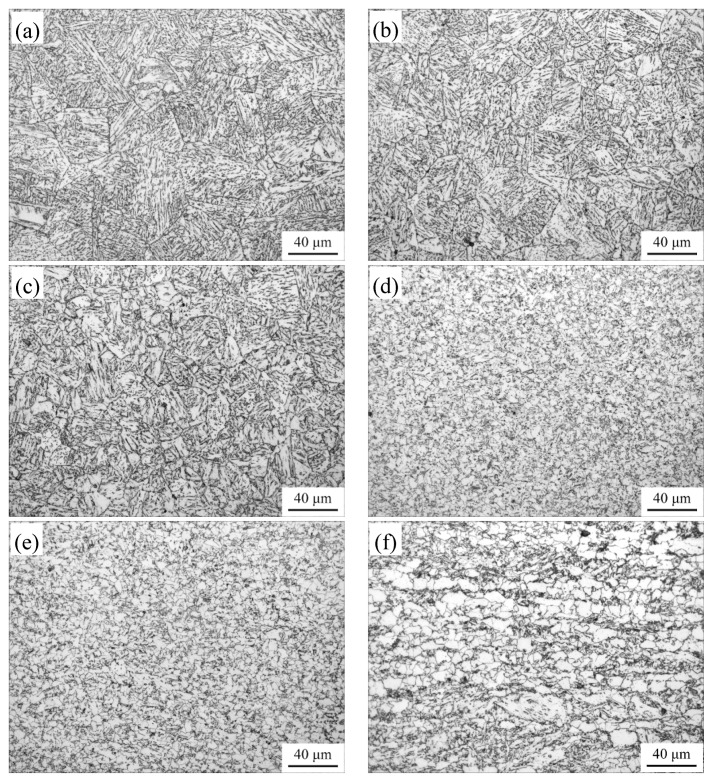
OM images of the simulated welded microstructures (**a**) 1350 °C; (**b**) 1250 °C; (**c**) 1150 °C; (**d**) 1050 °C; (**e**) 950 °C; (**f**) 850 °C.

**Figure 4 materials-12-01989-f004:**
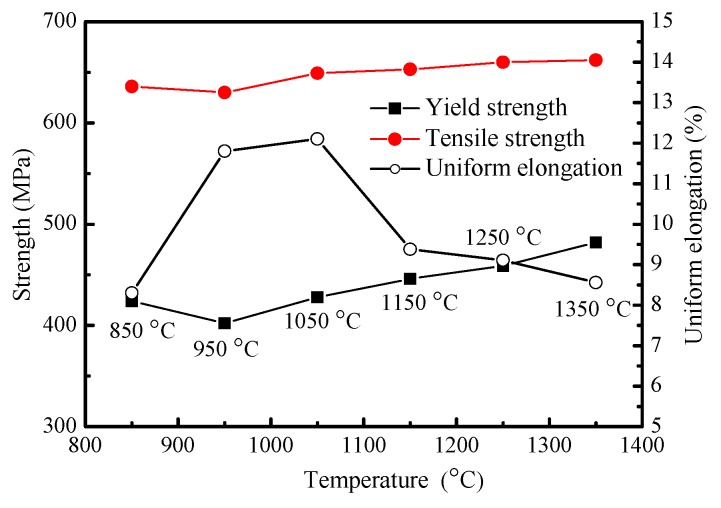
Tensile property under different peak temperatures.

**Figure 5 materials-12-01989-f005:**
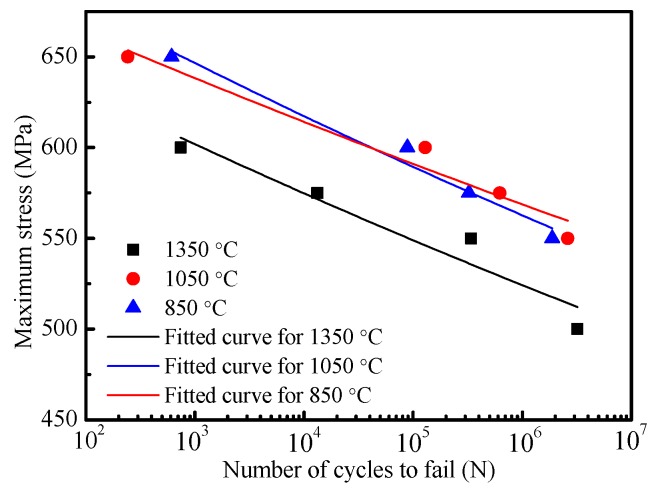
Fatigue life at different peak temperatures.

**Figure 6 materials-12-01989-f006:**
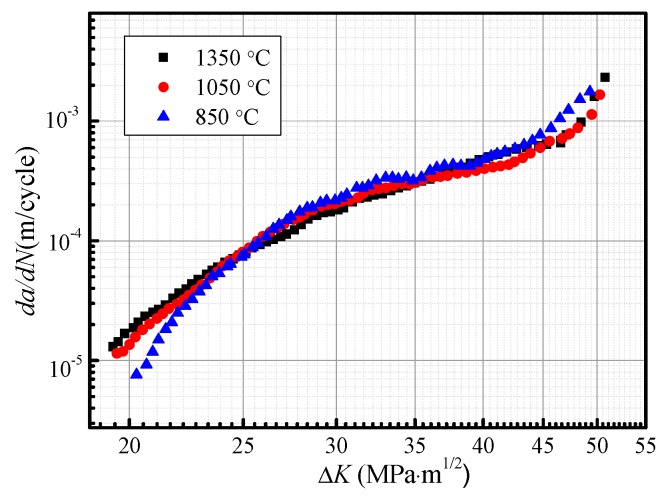
Fatigue crack growth rate as a function of Δ*K*.

**Figure 7 materials-12-01989-f007:**
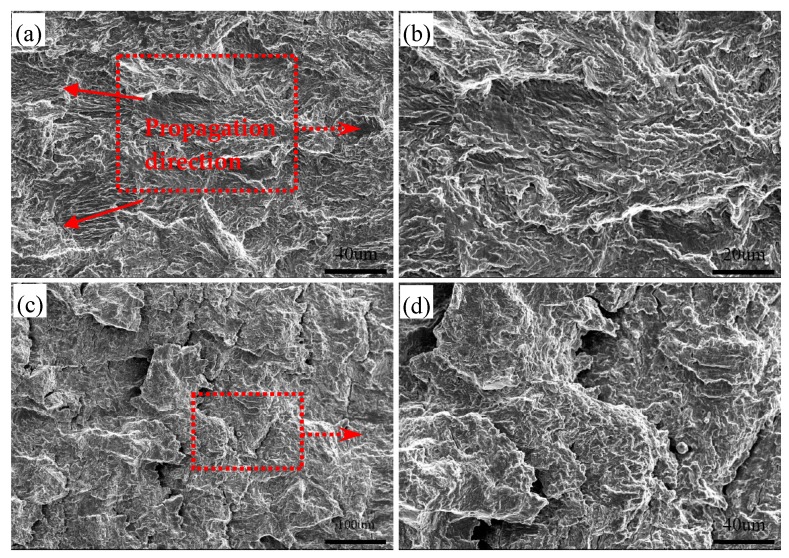
Typical SEM micrographs of fracture surface at peak temperature of 1350 °C. (**a**) the fracture surface in the early stage; (**b**) magnified view of the dashed box in (**a**); (**c**) the area far away from the crack source; (**d**) magnified view of the dashed box in (c).

**Figure 8 materials-12-01989-f008:**
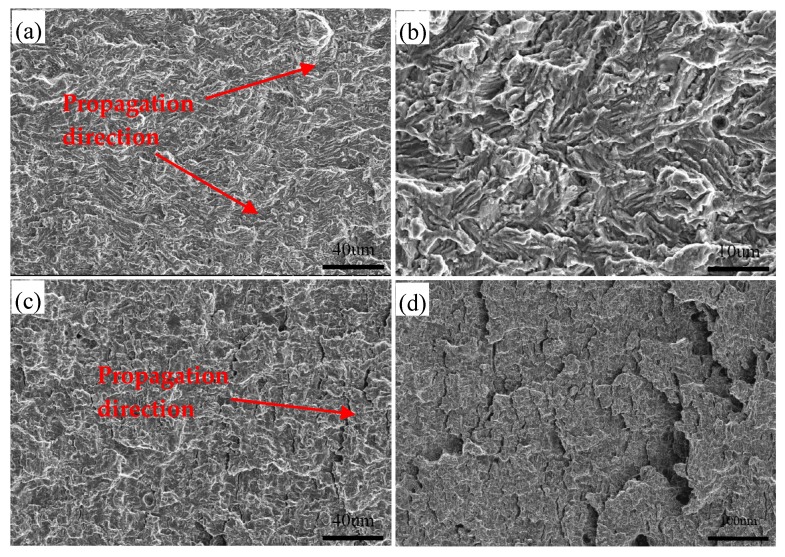
Typical SEM micrographs of fracture surface at peak temperature of 1050 °C. (**a**) the fracture surface in the early stage; (**b**) magnified view in (**a**); (**c**) the area far away from the crack source; (**d**) the area close to ultimate fracture region.

**Figure 9 materials-12-01989-f009:**
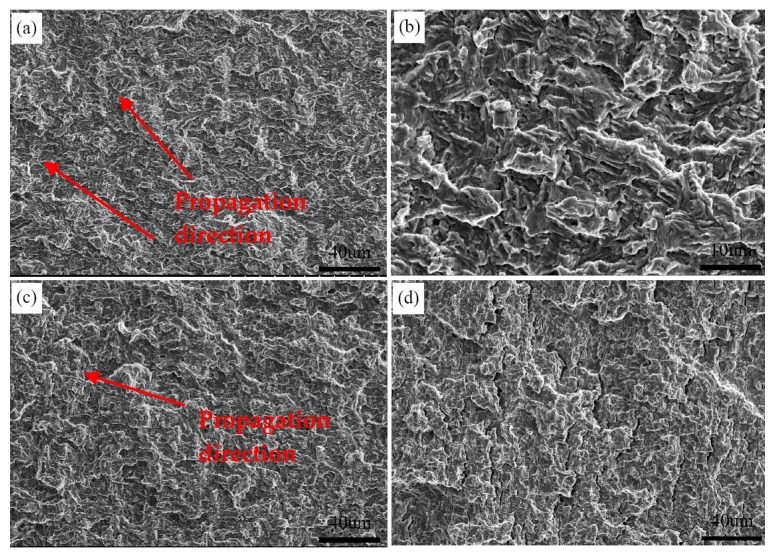
Typical SEM micrographs of fracture surface at peak temperature of 850 °C. (**a**) the fracture surface in the early stage; (**b**) magnified view in (**a**); (**c**) the area far away from the crack source; (**d**) the area close to ultimate fracture region.

**Figure 10 materials-12-01989-f010:**
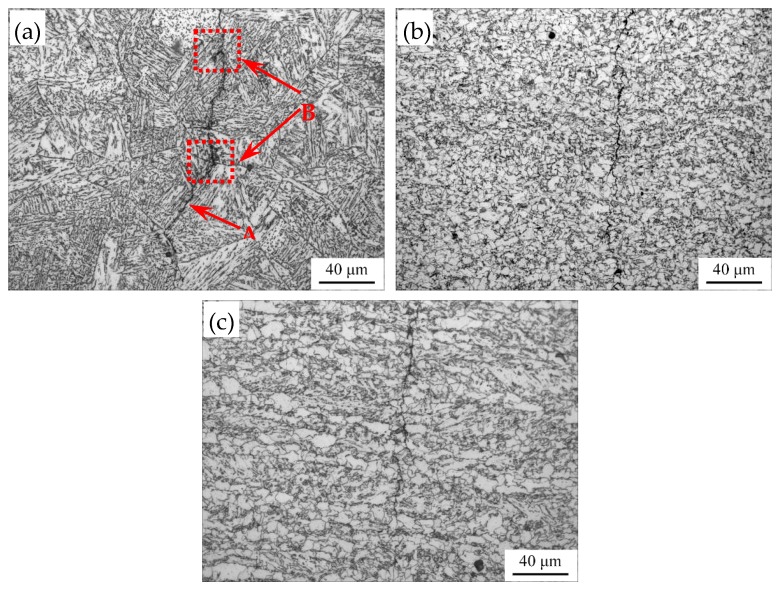
Typical OM micrographs of the fatigue crack path (**a**) 1350 °C; (**b**) 1050 °C; (**c**) 850 °C.

**Figure 11 materials-12-01989-f011:**
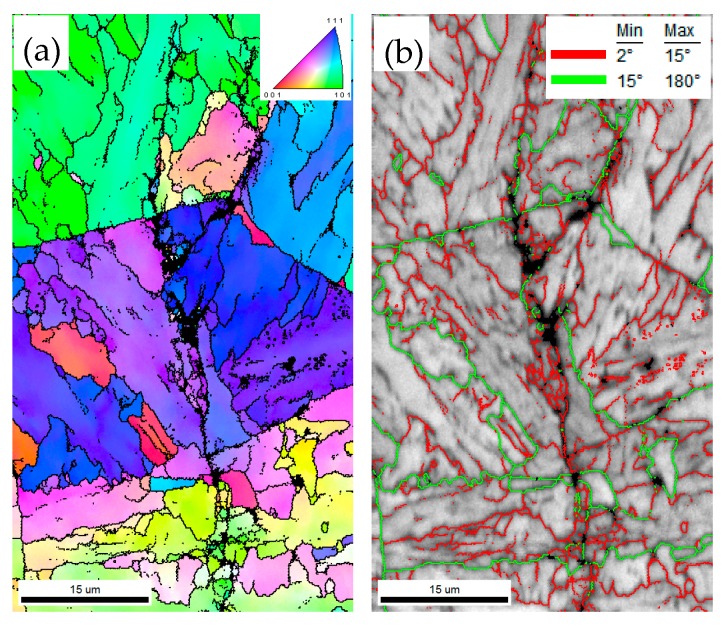
Crystallographic characteristics of the fatigue cracks at peak temperature of 1350 °C. (**a**) Orientation image maps; (**b**) Image quality maps with grain boundary misorientation distribution.

**Figure 12 materials-12-01989-f012:**
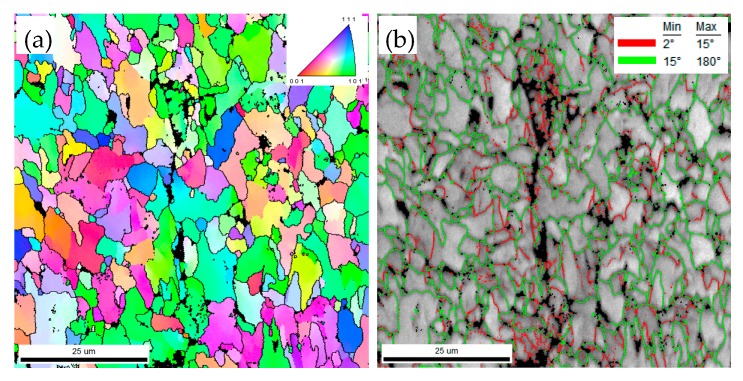
Crystallographic characteristics of the fatigue cracks at peak temperature of 1050 °C. (**a**) Orientation image maps; (**b**) Image quality maps with grain boundary misorientation distribution.

**Figure 13 materials-12-01989-f013:**
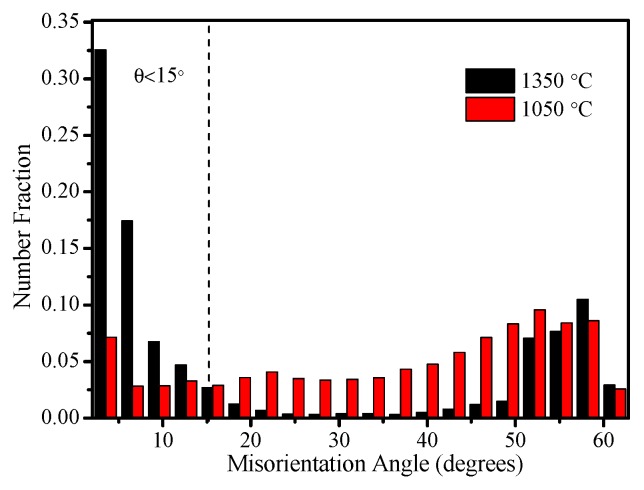
Number fraction of misoriented angles at different peak temperatures.

**Figure 14 materials-12-01989-f014:**
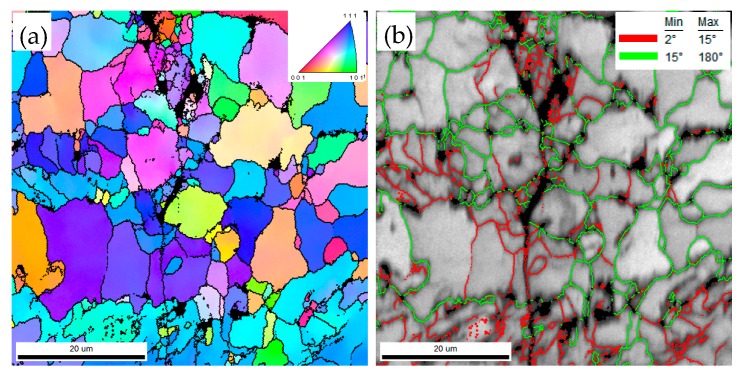
Crystallographic characteristics of the fatigue cracks at peak temperature of 850 °C. (**a**) Orientation image maps; (**b**) Image quality maps with grain boundary misorientation distribution.

**Figure 15 materials-12-01989-f015:**
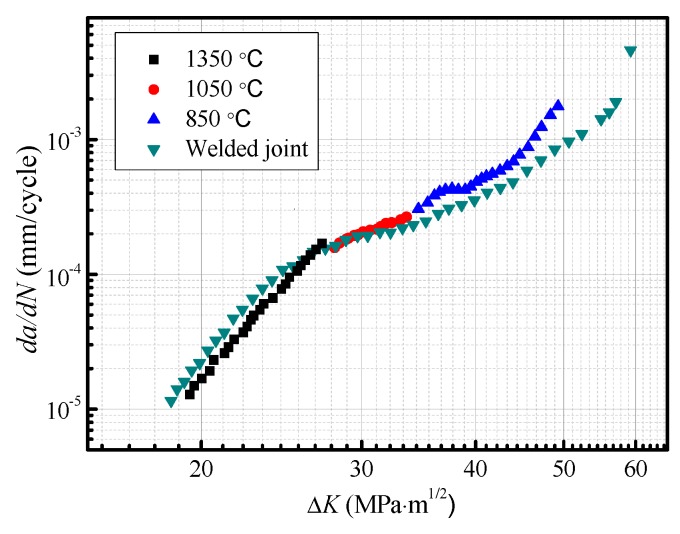
The difference between the actual welded joint specimen and thermal simulation specimen.

**Table 1 materials-12-01989-t001:** Chemical composition (in wt %) of the X80 steel selected in the present study.

C	Mn	Si	P	S	Mo	Ni	Cr	Cu	Nb	Ti	Al
0.07	1.69	0.21	0.011	0.002	0.003	0.255	0.223	0.129	0.082	0.015	0.035

**Table 2 materials-12-01989-t002:** Experimental projects and parameters in this study.

Experimental Project	Sample Type	Experiment Parameter
Weld thermal simulation test	Square specimen [Figure 2a]	Peak temperature: 1350 °C, 1250 °C, 1150 °C, 1050 °C, 950 °C and 850 °C
Fatigue life test	Stripe specimen [Figure 2c]	*R*: 0.1 Loading frequency: 20 Hz Loading waveform: sine wave
Fatigue crack propagation test	SENB specimen [Figure 2b]	*R*: 0.1 Loading frequency: 20 Hz Loading waveform: sine wave
Tensile test	Stripe specimen [Figure 2c]	Strain rate: 1 × 10^−3^ s^−1^

**Table 3 materials-12-01989-t003:** The different fitting parameters of the *da/dN*-Δ*K* curves between the welded joint specimen and thermal simulation sample.

HAZ		*m*	*C*
CGHAZ	Experiment	6.86	2.55 × 10^−14^
	Simulation	7.43	3.73 × 10^−15^
FGHAZ	Experiment	1.83	3.70 × 10^−7^
	Simulation	2.56	3.24 × 10^−8^
ICHAZ	Experiment	3.75	3.70 × 10^−10^
	Simulation	3.50	1.23 × 10^−9^

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
