# Peer review of "Characterization of Microstructures and Fatigue Properties for Dual-Phase Pipeline Steels by Gleeble Simulation of Heat-Affected Zone"

_materials, 2019, doi:10.3390/ma12121989_

Round 1

Reviewer 1 Report

The paper "Characterization of microstructures and fatigue properties for dual-phase pipeline steels by Gleeble simulation of heat-affected zone" describes the interesting results of the physical simulation of the welding process. It is known that the heating method in Gleeble may give very large difference in the temperature distribution in the sample. However authors did not give the details of the simulation experiments using Gleeble. Following comments also should be taken into account during uploading of the revision version:

1. Why did authors give thermal cycles results obtained only by FEM using ANSYS (Fig. 1b). It is better to give real temperatures which can be measured by thermocouple welded to the sample.

2. Authors apply energy density of 45 000 J/cm. However it is unclear what was the volume of the heat affected zone and what was the temperature distribution in this HAZ. It may has significant influence on the microstructure and mechanical properties.

Author Response

Point 1: Why did authors give thermal cycles results obtained only by FEM using ANSYS (Fig. 1b). It is better to give real temperatures which can be measured by thermocouple welded to the sample. 

Response 1: Thanks for your comments! Fig. 1b provides the testing program of weld thermal simulation test, which is determined by ANSYS analysis. This approach is also commonly used in many researches on welding simulation [1-3]. As you mentioned, it is better to give real temperatures measured by thermocouple welded to the sample. I completely agree with you on that. However, this method can be difficult to implement on-line. Therefore, considering the Application universality and feasibility, the ANSYS analysis was applied in this work. 

[1] X.W. Chen, G.Y. Qiao, Effects of Mo, Cr and Nb on microstructure and mechanical properties of heat affected zone for Nb-bearing X80 pipeline steels. Materials and Design 2014, 53, 888-901.

[2] Y. Ci, Z.Z. Zhang, Simulation study on heat-affected zone of high-strain X80 pipeline steel. Journal of Iron and Steel Research, International 2017, 21, 966-972.

[3] F. Mohammadi, F.F. Eliyan, A. Alfantazi, Corrosion of simulated weld HAZ of API X-80 pipeline steel. Corrosion Science, 2012, 63, 323-333.

Point 2: Authors apply energy density of 45 000 J/cm. However it is unclear what was the volume of the heat affected zone and what was the temperature distribution in this HAZ. It may has significant influence on the microstructure and mechanical properties.

Response 2: Thanks to you for your good comments. As you mentioned, the temperature distribution can have a significant impact on the microstructure and mechanical properties. Nevertheless, this paper mainly pay attention to the effect of microstructures in each HAZ sub-regions on fatigue properties. The microstructures in different sub-regions were obtained by changing the peak temperature. Although the temperature distribution in HAZ can change the size of each sub-region, there is no change in the microstructure. Therefore, only one heat input of thermal cycles was applied in this work and we take no account of the temperature distribution.

Reviewer 2 Report

The paper could be publised if the following changes are made:

- ¿Why some authors are from institution 1 and others form institution 2, but then 1 and 2 are the same institution? All authors must be under the same affilitaion in this case.

- Some Englhis review must be done in plurals, prepositions, etc. An English native speaker should revise the paper.

- Lines 83-84, the standard for tensile test is not indicated. The procedure followed and/or the standard should be included.

- Section 2. It is a bit confusing wich test and in wich conditions were carried out. A table detailling the test plan and result of each test should be included at the end of section 2.

- Figure 3 is divided in two different pages. The whole figure must be in the same page. The text should be modified in order for no to cut the figure.

- Figure 4. labels (a), (b), (c) …(f) of different microstructures would be wellcome in the figure in order to clarify it. (This is not mandatory, just recommended).

- Line 119, the terms CGHAZ, FGHAZ and ICHAZ should be explained in order to remember the reader what is each one, as their use becomes massive in the following of the test.

- Figure 5. For 850ºC and 1050ºC Máximum stresses of 650, 600, 575 and 550 Mpa were used, as for 1350ºC 600, 575, 550 and 500 MPa were employed. Why 500MPa was not used for 850 and 1050ºC and 650Mpa was not employed for 1350ºC? or were employed but not included in the graph? This must be justified and/or the values included, it looks a bit strange like this.

- Figure 5. The label of X axis should be "number of cycles to fail" instead of "number of cycles to fatige".

- Lines 127 - 132. Do really differences in the da/dn vs N graph among the three temperaturas are so big as commented? From my point of view in the three cases the behaviour is practically the same. Correct this, please.

- Figures 7, 8 and 9. It Will be very usefull to indicate the direction of propagatio in each one of the pictures, please.

- Lines 180 - 182. If says FGHAZ twice, is one of those incorrect? Wouldn' t it be CGHAZ or ICHAZ?

- Lines 203 - 205. This process of "divided and connected" looks confusing. It should be explained in a bit more of detail.

- Section "6. Patents". I thik this like should be elimintated. It was kept by mistake form the template, wasn't it?

Author Response

Point 1: Why some authors are from institution 1 and others form institution 2, but then 1 and 2 are the same institution? All authors must be under the same affilitaion in this case.

Response 1: Thank you for your careful work. We are very sorry for this mistake. We have carefully corrected it.

Point 2: Some English review must be done in plurals, prepositions, etc. An English native speaker should revise the paper.

Response 2: The language has been carefully checked and revised accordingly.

Point 3: Lines 83-84, the standard for tensile test is not indicated. The procedure followed and/or the standard should be included.

Response 3: Corresponding standard have been added in the revised manuscript as References [18].

Point 4: Section 2. It is a bit confusing wich test and in wich conditions were carried out. A table detailling the test plan and result of each test should be included at the end of section 2.

Response 4: Thanks to you for your good comments and I have added a table to describe experimental methods and conditions in the revised manuscript.

Table 2. Experimental projects and parameters in this study.

Experimental project Sample type Experiment parameter

Weld thermal simulation test Square specimen [Fig. 2(a)] Peak temperature: 1350 ℃, 1250 ℃, 1150 ℃, 1050 ℃, 950 ℃and 850 ℃

Fatigue life test Stripe specimen [Fig. 2(c)] R: 0.1

Loading frequency: 20 Hz

Loading waveform: sine wave

Fatigue crack propagation test SENB specimen [Fig. 2(b)] R: 0.1

Loading frequency: 20 Hz

Loading waveform: sine wave

Tensile test Stripe specimen [Fig. 2(c)] Strain rate: 1×10-3 s-1

The above contents have been added in the revised manuscript.

Point 5: Figure 3 is divided in two different pages. The whole figure must be in the same page. The text should be modified in order for no to cut the figure.

Response 5: Figure 3 has been kept in the same page.

Point 6: Figure 4. labels (a), (b), (c) …(f) of different microstructures would be wellcome in the figure in order to clarify it. (This is not mandatory, just recommended).

Response 6: Considering the Reviewer’s suggestion, we have marked different microstructures with labels in Figure 4.

Point 7: Line 119, the terms CGHAZ, FGHAZ and ICHAZ should be explained in order to remember the reader what is each one, as their use becomes massive in the following of the test.

Response 7: Thanks for your comments! We have re-written this part according to the Reviewer’s comments.

As described above, each HAZ sub-regions demonstrate many different microstructure characteristics because of different peak temperatures, especially CGHAZ (Fig. 3). Therefore, the results of fatigue life test may be related to the effect of microstructure on the crack propagation behavior.

The above contents have been added in the revised manuscript.

Point 8: Figure 5. For 850ºC and 1050ºC Máximum stresses of 650, 600, 575 and 550 Mpa were used, as for 1350ºC 600, 575, 550 and 500 MPa were employed. Why 500MPa was not used for 850 and 1050ºC and 650Mpa was not employed for 1350ºC? or were employed but not included in the graph? This must be justified and/or the values included, it looks a bit strange like this.

Response 8: Thanks to you for your good comments. The selection of maximum stress applied during fatigue life test depends on the magnitude of fatigue life at the last test. When the maximum stress is 550 MPa, 850 and 1050 ºC shows a high magnitude of fatigue life about 2106 cycles. This result can be used for comparative analysis with 1350 ºC effectively. In consideration of the test time, the maximum stress of 500 MPa was not employed for 850 and 1050 ºC. When the maximum stress is 600 MPa, 1350 ºC shows a minimum fatigue life about 700 cycles. Further increase in the maximum stress will have no effect on date analysis. Therefore, 650 Mpa was not employed for 1350 ºC in the test.

Point 9: Figure 5. The label of X axis should be "number of cycles to fail" instead of "number of cycles to fatige".

Response 9: We have carefully revised Figure 5 according to the Reviewer’s comments.

Point 10: Lines 127 - 132. Do really differences in the da/dn vs N graph among the three temperaturas are so big as commented? From my point of view in the three cases the behaviour is practically the same. Correct this, please.

Response 10: We are very sorry for our unclear description in Figure 6. We have re-written this part according to the Reviewer’s comments.

Point 11: Figures 7, 8 and 9. It Will be very usefull to indicate the direction of propagatio in each one of the pictures, please.

Response 11: The direction of propagation have been pointed out in Figures 7, 8 and 9.

Point 12: Lines 180 - 182. If says FGHAZ twice, is one of those incorrect? Wouldn' t it be CGHAZ or ICHAZ?

Response 12: Thank you for your careful work. We are very sorry for this mistake. The second FGHAZ should be replace with CGHAZ, and We have corrected it in the revised manuscript.

Point 13: Lines 203 - 205. This process of "divided and connected" looks confusing. It should be explained in a bit more of detail.

Response 13: We have re-written this part according to the Reviewer’s comments.

To compare and analyze the difference between simulated and actual welded joints in fatigue crack growth behavior, a da/dN-K curve after mosaic is obtained by thermal simulation samples, as shown in Figure 15. The simulated curve consists of different stages of each da/dN-K curves in HAZ sub-regions.

The above contents have been added in the revised manuscript.

Point 14: Section "6. Patents". I thik this like should be elimintated. It was kept by mistake form the template, wasn't it?

Response 14: We are very sorry for this mistake. The corresponding contents has been eliminated in the revised manuscript.

Round 2

Reviewer 1 Report

The article may be accepted for publication.